# Highly Sensitive Multichannel Fano Resonance-Based Plasmonic Sensor for Refractive Index and Temperature Sensing Application

**Chung-Ting Chou Chao [1],\* and Yuan-Fong Chou Chau [2],\***

1 Department of Optoelectronics and Materials Technology, National Taiwan Ocean University, Keelung 20224, Taiwan
2 Centre for Advanced Material and Energy Sciences, Universiti Brunei Darussalam, Tungku Link, Bandar Seri Begawan BE1410, Brunei
* Correspondence: 10889013@email.ntou.edu.tw (C.-T.C.C.); chou.fong@ubd.edu.bn (Y.-F.C.C.)

**Abstract:** We propose a susceptible multichannel plasmonic sensor for sensing refractive index (RI) and temperature media working in the visible to near-infrared range. The proposed structure's resonator consists of an elliptical-shaped ring with two stubs at two sides and four metal nanorods side-coupled to two separated metal–insulator–metal waveguides. The optical responses of the structure, including transmittance spectra and magnetic and electric field distributions, are investigated using the finite element method (FEM) to obtain the optimal structural parameters. The designed structure supports five channels of Fano resonance modes because of the interaction between the narrowband mode of the elliptical-shaped ring resonator and the broadband mode of two separated MIM WGs. The maximum sensitivity values can reach 4500 nm/RIU for RI sensing, and the temperature sensitivity can get 1.00 nm/°C. The designed device exhibits excellent sensing performance and could pave the way for sensing devices with significantly higher sensitivity.

**Keywords:** multichannel sensor; metal–insulator–metal; temperature sensitivity; Fano resonances

## 1. Introduction

Surface plasmon polaritons (SPPs) can achieve high-impact optical devices due to their optical characteristics of overcoming the typical diffraction limit in the nanometer dimension. It is a promising research field in optical manipulation, integration, processing, and sensing [1–7]. The electromagnetic (EM) wave can generate by inducing the sensing device through SPPs waves that are highly localized on the metal–dielectric interface. Metal–insulator–metal (MIM) waveguide (WG) is an SPPs-based nanostructure consisting of a dielectric (or air) layer placed in between two metals, making up of MIM WG coupled to a resonator (or cavity) [8–11]. Due to the unique properties of SPPs, e.g., available wavelength range, high confinement, low loss, and easy fabrication, many research groups focus their attention on the applications of MIM WGs [12,13]. Plasmonic refractive index (RI) sensors based on MIM WG have attracted considerable attention for their unique property for molecular binding and label-free detection [14–16].

Plasmonic MIM WG-based RI and temperature detection sensors are excellent options for achieving precise and sensitive biosensors. The coupling effects between the MIM WG and the resonator are crucially dependent on the aspect of the resonator. To date, several MIM WGs with diverse shapes of resonators have been investigated, such as circular/rectangular-ring shaped [17,18], triangular/bowtie-shaped [19,20], tooth-shaped [21,22], H-shaped [23], r-shaped [24], U-shaped [25], T-type [26], M-shaped [27], cross-shaped [28], elliptical-shaped, etc. The elliptical-shaped resonator merits simplicity and controllable resonance wavelengths ($\lambda_{res}$) [29]. Therefore, different elliptical-shaped resonators have been designed and developed. Zafar et al. [30] created a RI sensor that used two elliptical-ring-shaped resonators and

obtained 1100 nm/RIU sensitivity. Salah et al. [31] proposed an elliptical-like racetrack resonator, attaining a RI sensitivity of 1400 nm/RIU. In [32], Jun et al. proposed an elliptical-like resonator and claimed a maximum sensitivity of 1261 nm/RIU.

MIM WG-coupled resonators can offer unique optical properties, e.g., Fano resonance [33]. Fano resonance reveals asymmetric, sharp spectral profiles and substantial light field enhancements [34]. The resonator functions as a continuous spectral state generator, while the MIM WG exhibits a discrete spectral state generation [35]. The Fano resonance phenomenon is attributed to the destructive interference of the discrete and continuous state [33,36–40].

Chen et al. developed a Fano resonance-based temperature sensor with MIM WG side-coupled to two circular-ring-shaped resonators. They claimed that the Fano resonance could also be tuned by the ethanol's temperature [41]. Yu et al. designed a temperature sensor based on a racetrack resonant cavity, with the RI and temperature sensitivity of 1503.7 nm/RIU and 0.75 nm/°C, respectively [42]. Lin et al. used two WGs coupled with a symmetry-breaking ring resonator to design a Fano resonance-based temperature sensor and obtained the sensitivity to temperature of ethanol analyte achieving 0.701 nm/°C [43].

This paper aims to design a Fano resonance-based multichannel RI and temperature sensor structure comprised of a MIM WG side-coupled to an elliptical-ring-shaped resonator. The resonator includes four Ag nanorods and two air stubs at two ends of the resonator. Using four metal nanorods and stub-shaped cavities at two ends of the resonator could enhance the gap plasmon resonance (GPR) and cavity plasmon resonance (CPR) in the MIM WG-based sensing system. We investigated and compared four cases (i.e., cases 1–4) of sensor structures. The numerical simulations were performed using the finite element method (FEM) with perfectly matched and scattering boundary conditions to prevent the reflected light on the boundary interface. We investigated the transmittance spectrum, magnetic and electric field distributions, and structural parameters. Simulation results show that the designed case 4 structure can induce five Fano resonance modes. The proposed case 4 structure can obtain high RI and temperature sensitivity by optimizing these structural parameters. The key finding of this work is that the existence of four silver nanorod defects in the resonator has a great influence on the sensitivity performance, which offers an additional degree to tune the optical response of the plasmonic system. The advantage of the proposed structure is that it has a compact size of $5.35 \times 10^4$ nm$^2$, and can offer five channels of Fano resonance modes, which is superior to the reported articles having only one working wavelength (e.g., [44]).

The organization of this article is as follows: Section 2 introduces the simulation models of cases 1–4 structures. Besides, we illustrate the simulation method, fundamental concept, and basic formula. Section 3 compares the Fano resonance peaks of cases 1–4 structures and demonstrates that the case 4 structure is an optimal design with five Fano resonance peaks and acceptable full width at half maximum (FWHM). Furthermore, we vary the structural parameters to optimize the geometrical parameters. In Section 4, we use the proposed case 4 structure to inspect the temperature sensing performance and compare the sensing performance with previously published papers. Finally, we summarized our work at the end of this paper.

## 2. Simulation Models, Methods, and Basic Formulas

Figure 1a–d show the top view of the designed cases 1–4 of the plasmonic optical sensor. Case 1 comprises only two separated MIM WGs (Figure 1a). Case 2 is the same as case 1 but is side-coupled to a resonator, i.e., an elliptical-ring-shaped resonator (Figure 1b). Case 3 is based on case 2, including four Ag nanorods placed at the elliptical-ring-shaped resonator's top, bottom, left, and right ends (Figure 1c). The proposed entire structure (i.e., case 4) is based on case 3, which connects with two stubs (Figure 1d) at the left and right ends of the elliptical-ring-shaped resonator. The structural parameters are as follows: the width of MIM WGs and elliptical-ring-shaped resonator (*w*), the separated distance between two separated MIM WGs (*d*), the gap between the MIM WGs and the resonator

(*g*), the stub's width (*a*), the stub's length (*b*), the outer semi-major axis and semi-minor axis of the elliptical-ring-shaped resonator ($R_x$ and $R_y$), and the radius of Ag nanorods (*r*), respectively.

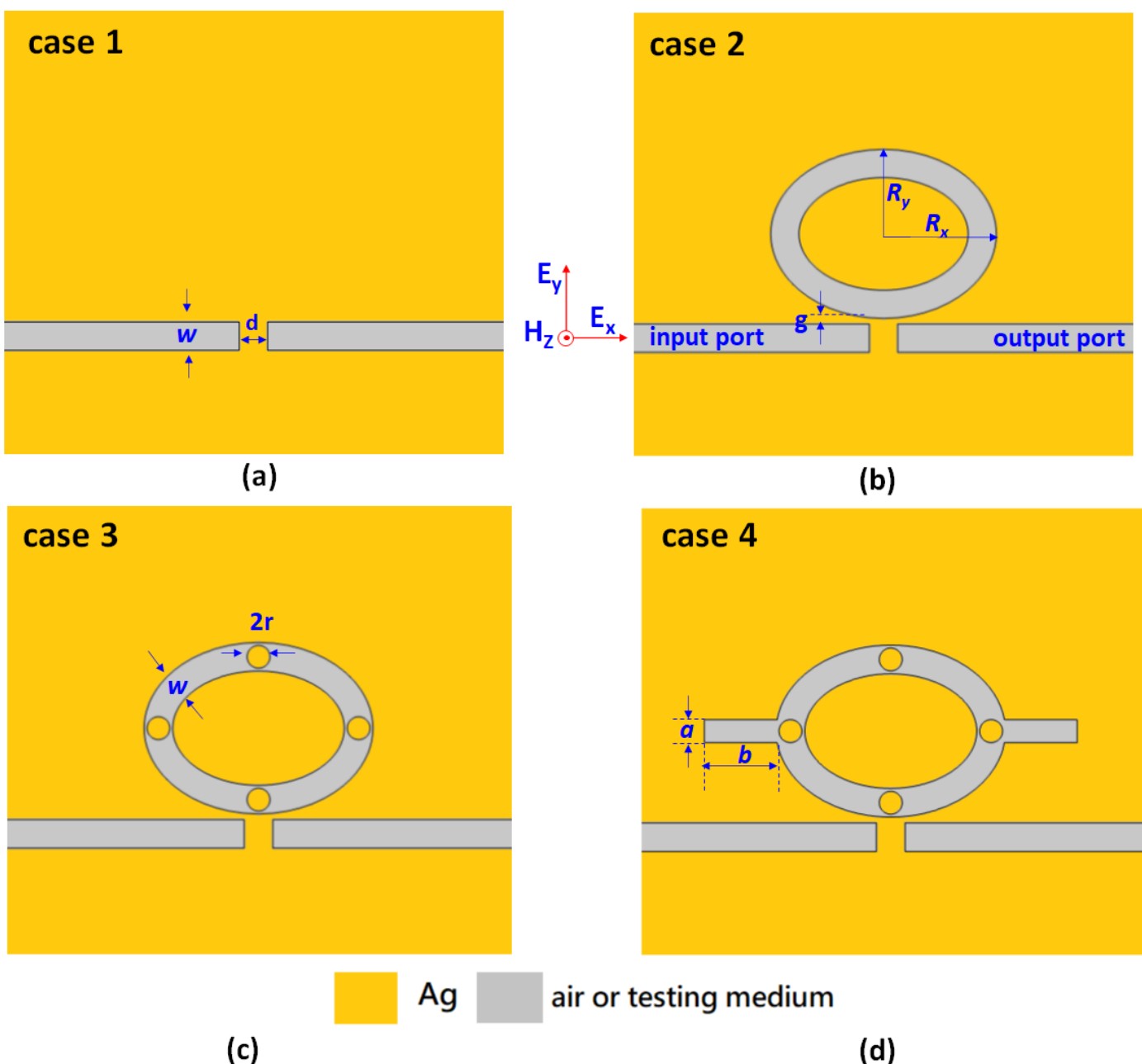

**Figure 1.** Top view of the designed MIM plasmonic sensors: (**a**) case 1, (**b**) case 2, (**c**) case 3, and (**d**) case 4, respectively. Where $H_z$, $E_y$ and $E_x$ represent the magnetic field component along z-axis, electric field component along y-axis and electric field component along x-axis, respectively.

We calculate the transmittance spectrum and EM field distribution using FEM-based commercial software COMSOL Multiphysics. Considering that the z-axis is much thicker than the x–y plane, we use a two-dimensional simulation model instead of a three-dimensional model to save computer resources [45]. A TM-polarized incident light coupled with the fundamental SPP mode from the MIM WG's input port to the output port. We use Ag as the plasmonic material because it can ensure high-field intensity and lower power consumption. The Drude model can express the permittivity data of Ag [46].

The SPPs can form in the plasmonic system when the incident EM wave satisfies $\Delta\varphi = 2\pi N$ ($N$ is an integer), the $\lambda_{res}$ can describe as [47,48].

$$\lambda_{res} = \frac{2L_{eff}Re(n_{\text{eff}})}{N - \frac{\varphi}{2\pi}} \ (N = 1, 2, 3\ldots) \tag{1}$$

$L_{\text{eff}}$, $\varphi$, and $Re(n_{\text{eff}})$ represent the resonator's effective length, phase shift, and the effective RI's real part. The $Re(n_{\text{eff}})$ can determine by Equation (2) [49].

$$Re(n_{\text{eff}}) = \left(\varepsilon_{Ag} + \left(\frac{k}{k_0}\right)^2\right)^{\frac{1}{2}} \tag{2}$$

In Equation (2), $k = 2\pi/\lambda$ stands for the wave vector, and $k_0$ denotes the free space wave vector.

The designed structure can serve as a temperature sensor. A liquid, ethanol, with a high RI temperature coefficient (i.e., $dn/dT = 3.94 \times 10^{-4}$) can infiltrate the MIM WGs and resonator. Ethanol's RI can be expressed as [50]:

$$n = 1.36048 - 3.94 \times 10^{-4} \ (T - T_0) \tag{3}$$

Here $T_0 = 20\,°\text{C}$ (room temperature), and $T$ is the ambient temperature. When ethanol is employed as the measuring liquid, the sensor's operating temperature range is between $-114.3$ and $78\,°\text{C}$ [51]. Table 1 shows the basic formulas used in this work.

**Table 1.** Basic formulas used in this work.

| Mane | Formula | Unit |
|---|---|---|
| $T$ (transmittance) [52] | $T = P_{out}/P_{in}$ | |
| $S$ (sensitivity) [53] | $S = \Delta\lambda/\Delta n$ | nm/RIU |
| $FOM$ (figure of merit) [54] | $FOM = S/FWHM$ | 1/RIU |
| $QF$ (quality factor) [55] | $QF = \lambda_{res}/FWHM$ | |
| $\Delta D$ (dipping strength) [56] | $\Delta D = (T_{max} - T_{min}) \times 100\%$ | |
| $S_T$ (temperature sensitivity)[57] | $S_T = \Delta n/\Delta T$ | nm/°C |

Where $P_{out}$, $P_{in}$, $\Delta\lambda$, and $\Delta n$ are output power, input power, $\lambda_{res}$ shift, and the difference in the RI. Besides, $T_{max}$ and $T_{min}$ signify the maximum and minimum transmittance, and the $P_{out}$ (output power) and $P_{in}$ (input power) can obtain from integral values of energy-flux density.

## 3. Optimization of the Geometrical Parameters

Table 2 lists the initial geometrical parameters. We first investigate the transmittance spectrum of the SPPs mode for the proposed cases 1–4 structures. Figure 2a displays the transmittance spectrum of the designed plasmonic sensors (i.e., cases 1–4) in the wavelength range of 500–1800 nm. In Figure 2a, the black case 1 curve shows a low broadband mode transmittance and decays as the increasing incident wavelength due to the two disconnected MIM WGs blocked by the silver with a distance of $d = 25$ nm. However, when case 1 couples with different configurations of an elliptical-ring-shaped resonator (i.e., cases 2–4), the transmittance spectra reveal hybridization of broadband and narrowband modes and result in other numbers of Fano resonance peaks. Thus, it demonstrates that cases 2–4 structures can produce Fano resonance modes. The mode number is closely related to the different resonance conditions between the MIM WGs and the resonator. As seen in the red curve of case 2, two transmittance peaks correspond to Fano resonance modes at $\lambda_{res} = 1325$ nm (mode 1) and $\lambda_{res} = 680$ nm (mode 2). Case 3 (pink curve) generates three Fano resonance modes at $\lambda_{res} = 1605$ nm (mode 1), $\lambda_{res} = 950$ nm (mode 2), $\lambda_{res} = 685$ nm (mode 3), and $\lambda_{res} = 555$ nm (mode 4), respectively. Compared to case 2, the two additional Fano resonance peaks obtained from case 3 attributed to the existence of four Ag nanorods in the elliptical-ring-shaped resonator. In the entire structure (case 4), there are five Fano resonance modes at $\lambda_{res} = 1501$ nm (mode 1), 1090 nm (mode 2),

915 nm (mode 3), 600 nm (mode 4), and 535 nm (mode 5), respectively. It is worth noting that mode 2 in case 3 splits into mode 2 and mode 3 in case 4, showing that the two stubs in case 4 can mediate the coupling effect in the resonator. Besides, the average value of $\Delta D$ at modes 1–2 for case 2, modes 1–4 for case 3, and modes 1–5 for case 4 are 59.16%, 42.49%, and 48.45%, indicating that the case 4 structure has both merits of more channels of resonance modes and a good coupling effect between the MIM WGs and resonator.

**Table 2.** Initial geometrical parameters of the proposed plasmonic sensor.

| $w$ (nm) | $R_x$ (nm) | $R_y$ (nm) | $g$ (nm) | $a$ (nm) | $b$ (nm) | $d$ (nm) | $r$ (nm) |
|---|---|---|---|---|---|---|---|
| 50 | 200 | $R_x$-50 | 10 | 40 | 160 | 25 | 20 |

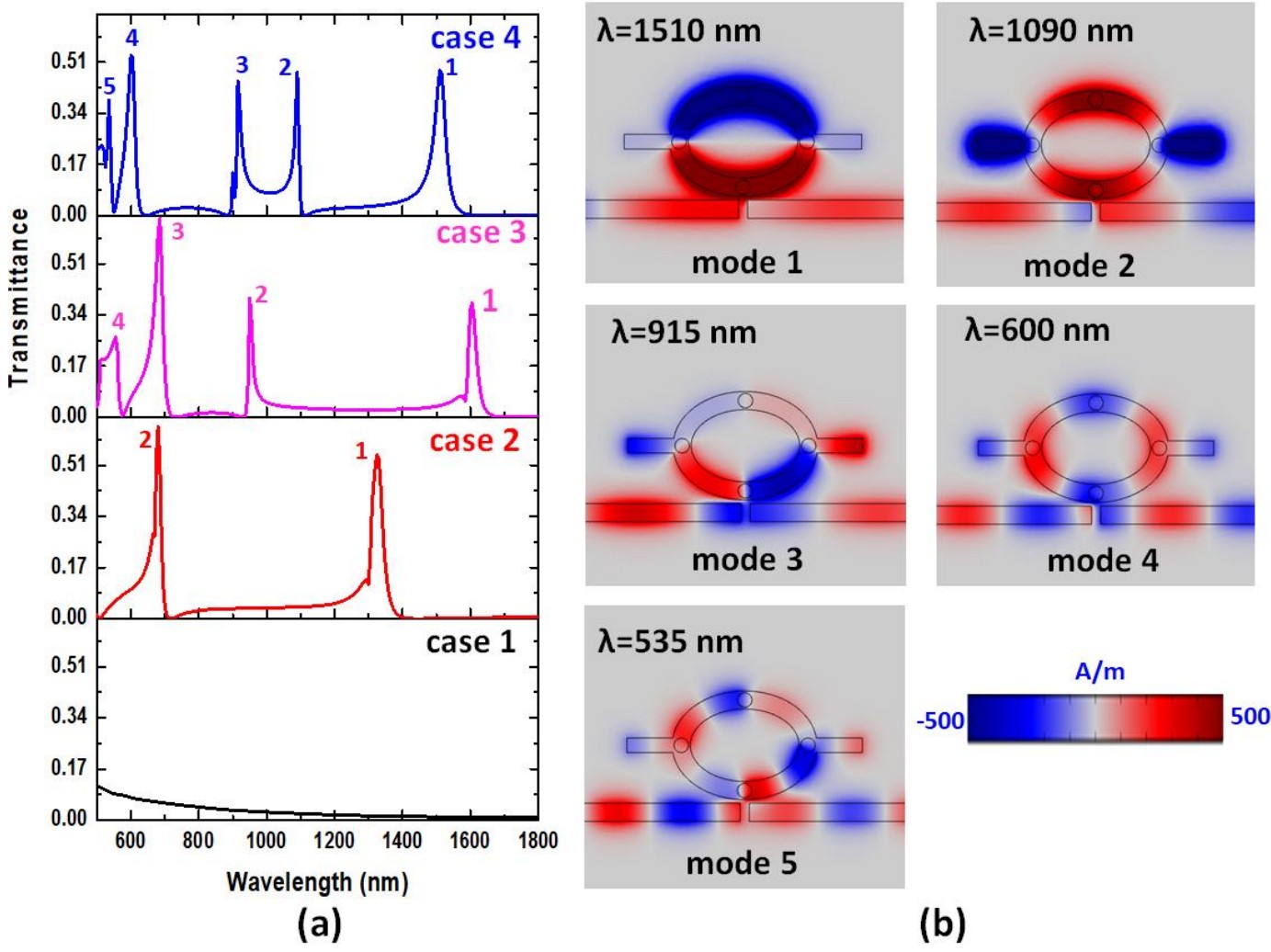

**Figure 2.** (**a**) Transmittance spectrum of the designed plasmonic sensors (i.e., cases 1–4) in the wavelength range of 500–1800 nm. (**b**) Truncate views of magnetic field (H-field) distributions at corresponding $\lambda_{res}$ of mode 1 to 5 of case 4 structure. Numbers mark the valid resonance modes.

In Figure 2a, we found that the resonator's patterns are vital in offering more plasmon resonance modes depending on CPR and GPR. These Fano resonance modes originate from CPR since the coupling effect between the MIM WGs and resonator. At the same time, the Fano resonance modes generated from GPR are due to the gap between Ag nanorods and the resonator's walls. To understand the mechanism of the resulting Fano resonance modes in the designed case 4 structure, Figure 2b plots the H-field distributions at corresponding $\lambda_{res}$ of modes 1 to 5, respectively. As observed, the different phases are associated with the

$\lambda_{res}$ and show the number of H-field petals in the resonator. The petal number of H-field decreases with the increase of $\lambda_{res}$ due to the longer optical path experiences by the larger wavelength. When the incident wavelength is at resonance modes, the H-fields are well confined in the resonator, exhibiting a significant CPR effect [58,59]. The H-field intensity is more prominent in the different parts of the resonator and shows an excellent coupling effect. Besides, the dipole effect, arising from the surface of Ag nanorods and the walls of MIM WGs and elliptic-ring-shaped resonators, forms the vigorous confinement of SPPs and supports constructive interference in the resonator [60]. Therefore, the prominent Fano resonance peaks can be induced, as shown in Figure 2a.

Subsequently, we inspect the influence on the shape of the Fano resonance curve by varying the parameters of the case 4 structure, i.e., *g, d, a, b, r,* and *R,* respectively. Figure 3a,b show the transmittance spectrum of *g* = (5, 10, 15, 20, 25, 30) nm and *d* = (0, 5, 15, 25, 35, 50) nm, respectively. Following Table 2, the value of *d* is 25 nm when the value of *g* increases from 5 to 30 nm, and the value of *g* is 10 nm when the value of *d* increases from 0 to 50 nm. The coupling effect between the MIM WGs and resonator is closely related to the *g* size. The shorter *g* is, the more vital interaction will be happening. As shown in Figure 3a, The dipping strength $\Delta D$ decreases with the increasing *g* due to the lesser coupling effect. We found that the case of *g* = 5 nm has the highest $\Delta D$ (i.e., 67.76%, 66.44%, 71.45%, and 70.74% for modes 1–4, respectively) compared to other values of *g*. However, it reveals a larger FWHM because a shorter *g* induces a strong interaction between the MIM WGs and the resonator. We choose *g* = 10 nm as the optimal value in the viewpoint of smaller FWHM and the acceptable $\Delta D$. Figure 3b illustrates the transmittance spectrum of d values ranging from 0–50 nm. When *d* = 0 nm, i.e., a MIM WG, the transmittance curve presents five dips with a Lorentz-like resonance. When distance d blocks the two MIM WGs, the Fano resonance peak appears, and the $\Delta D$ and FWHM decrease with the increasing *d*. In Figure 3b, the available range of *d* is 15–50 nm based on the depth of $\Delta D$ and the width of FWHM.

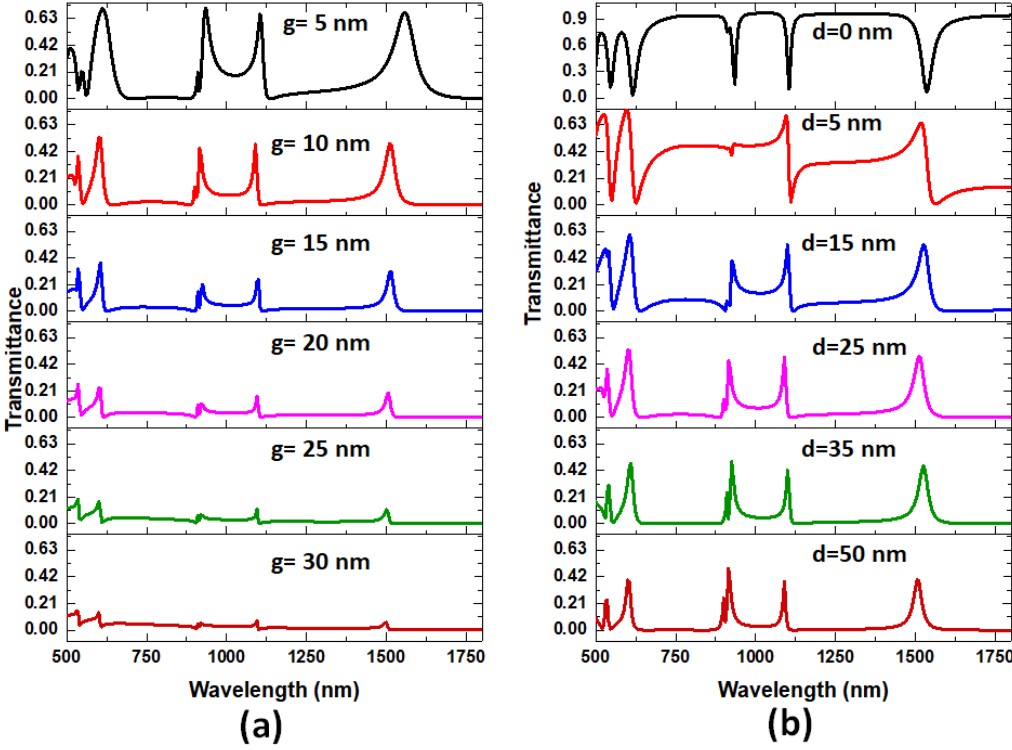

**Figure 3.** Transmittance spectrum of case 4 by varying (**a**) *g* = (5, 10, 15, 20, 25, 30) nm, and (**b**) *d* = (0, 5, 15, 25, 35, 50) nm, respectively.

As shown in the H-field distribution in Figure 2b, the two stubs can mediate the coupling effect in the resonator. It means that the Fano resonance peak can be manipulated by a different area of the stub (i.e., a × b). Figure 4a,b exhibits the transmittance spectrum of $a$ = (10, 20, 30, 40, 50, 60, 70, 80) nm, and $d$ = (0, 100, 150, 200, 250, 300, 350, 400) nm, respectively. We used the other structural parameters in Table 2. The values of $b$ and $d$ are 160 and 25 nm when the value of $a$ increases from 10 to 80 nm, and the values of $a$ and $b$ are 40 and 160 nm when the value of $d$ increases from 0 to 40 nm. Both $a$ and $b$ of mode 1 reveal a blueshift because of the decrease of effective RI ($n_{eff}$, see Equation (1)) of the whole plasmonic system with a larger area of stubs. We notice that the resonance peak (i.e., $\Delta D$) of mode 2 increases with a rising $a$ (see Figure 4a), while the $\Delta D$ of mode 2 increases with the promoting $b$, reaches its highest peak when $b$ = 300 nm, and then decreases with a higher value of $b$ (see Figure 4b). The acceptable range of $a$ and $b$ is broad and can select $a$ = 10–80 nm and $b$ = 0–400 nm, depending on the desired operating wavelength.

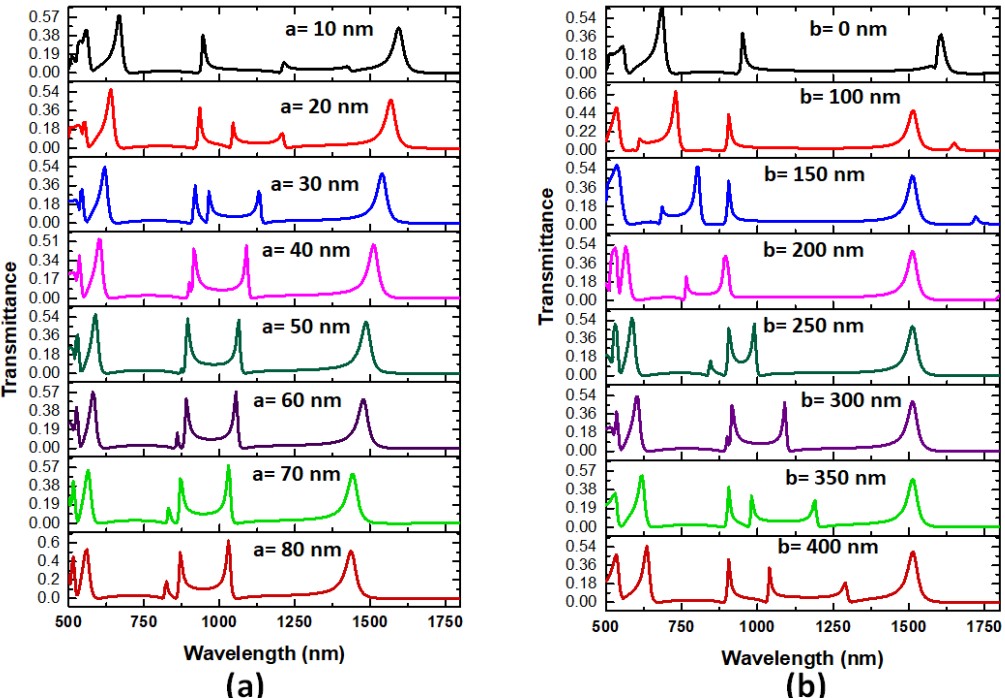

**Figure 4.** Transmittance spectrum of case 4 by varying (**a**) $a$ = (10, 20, 30, 40, 50, 60, 70, 80) nm, and (**b**) $d$ = (0, 100, 150, 200, 250, 300, 350, 400) nm, respectively.

The size of Ag nanorods and the elliptical-ring resonator's radius can influence the resonance wavelength position. According to Equation (1), the former will change the effective RI ($n_{eff}$), and the latter will affect the effective length of the resonator ($L_{eff}$). Figure 5a,b illustrates the transmittance spectrum of $r$ = (0, 5, 10, 15, 20, 22) nm and $R_x$ = (150, 175, 200, 225, 250, 300) nm, respectively. The other structural parameters are used the same as in Table 2. As expected, the Fano resonance peaks redshift with the increase of $r$ (from $\lambda_{res}$ = 1325–1600 nm) and $R$ (from $\lambda_{res}$ = 1095–2600 nm). The available range of $r$ and $R_x$ can be chosen in the range of $r$ = 0–22 nm and $R_x$ = 150–300 nm, respectively. We can explain these results because of the difference in matching impedance conditions between the resonator and bus waveguide. The change of $r$ and $R$ results in varying impedance when the resonance condition in the resonator is satisfied; meanwhile, $\lambda_{res}$ should be increased to guarantee impedance between the bus waveguide and the resonator when the effective RI ($n_{eff}$) of the resonator region is changed by $r$ and $R$ [61].

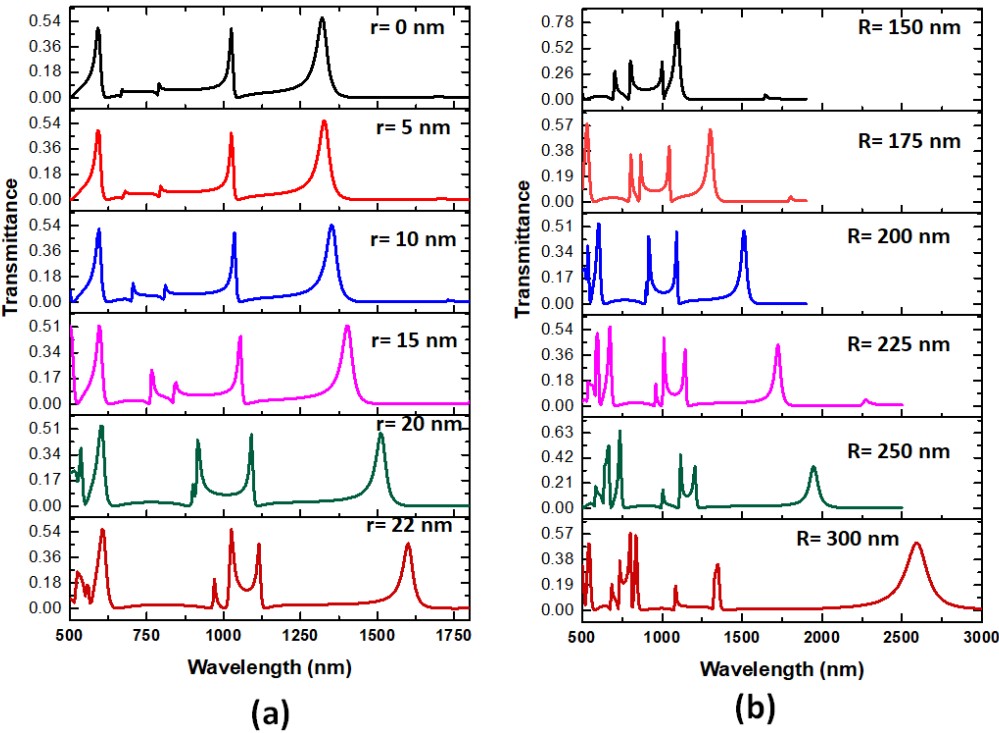

**Figure 5.** Transmittance spectrum of case 4 by varying (**a**) $r$ = (0, 5, 10, 15, 20, 22) nm, and (**b**) $R_x$ = (150, 175, 200, 225, 250, 300) nm, respectively.

The dimensions of the proposed structure are in nanometer order, and the radius of Ag nanorods ($r$) is 0~22 nm. It is a challenge to manufacture and measure a structure with such small dimensions. If the $r$ is too small (e.g., $r$ = 5 nm) or too big (e.g., $r$ = 22 nm), it causes difficulty to fabricate the sensor. The fabrication of the proposed sensor is straightforward. Thanks to the advancement in the fabrication of nanophotonic devices, the proposed sensor can be manufactured using stripping and electric beam lithography (EBL) processes [62], which can imprint custom pictures with sub-10 nm resolution [63]. A thin layer of Ag can be deposited on a silica substrate followed by EBL patterning. We can use wet etching (e.g., diluted nitric acid) to clear the unwanted Ag and form the Ag nanorods in the right positions through the lift-off technique.

For inspecting the sensing performance in mode 1 by varying the different $R$ of the case 4 structure, Figure 6 shows the transmittance spectrum of case 4 by changing $R_x$ in the range of $R_x$ = (150, 175, 200, 225, 250, 300) nm and the RI ($n$) of surrounding material changing from 1.00 to 1.01. Table 3 compares the $S$, FOM, $Q$-factor, and $\Delta D$ of case 4 with different $R_x$ for mode 1. The value of $S$, FOM, $Q$-factor, and $\Delta D$ can reach 4500.00 nm/RIU, 34.62 RIU$^{-1}$, 19.92, and 51.43% for $R_x$ = 300 nm. The case 4 structure can be used as a high-precision plasmonic RI sensor for $n$ in the interval range of 0.01.

**Table 3.** The S, FOM, Q-factor, and ΔD of case 4 with different $R$ for mode 1.

| $R_x$ (nm) | 150 | 175 | 200 | 225 | 250 | 275 | 300 |
|---|---|---|---|---|---|---|---|
| S (nm/RIU) | 1000 | 1500 | 1500 | 1500 | 3000 | 4000 | 4500 |
| FOM (1/RIU) | 100.00 | 150.00 | 150.00 | 150.00 | 200.00 | 40.00 | 34.62 |
| Q-factor | 109.50 | 130.00 | 151.00 | 172.50 | 129.67 | 22.25 | 19.92 |
| ΔD (%) | 78.54 | 54.07 | 48.43 | 42.86 | 35.99 | 43.22 | 51.43 |

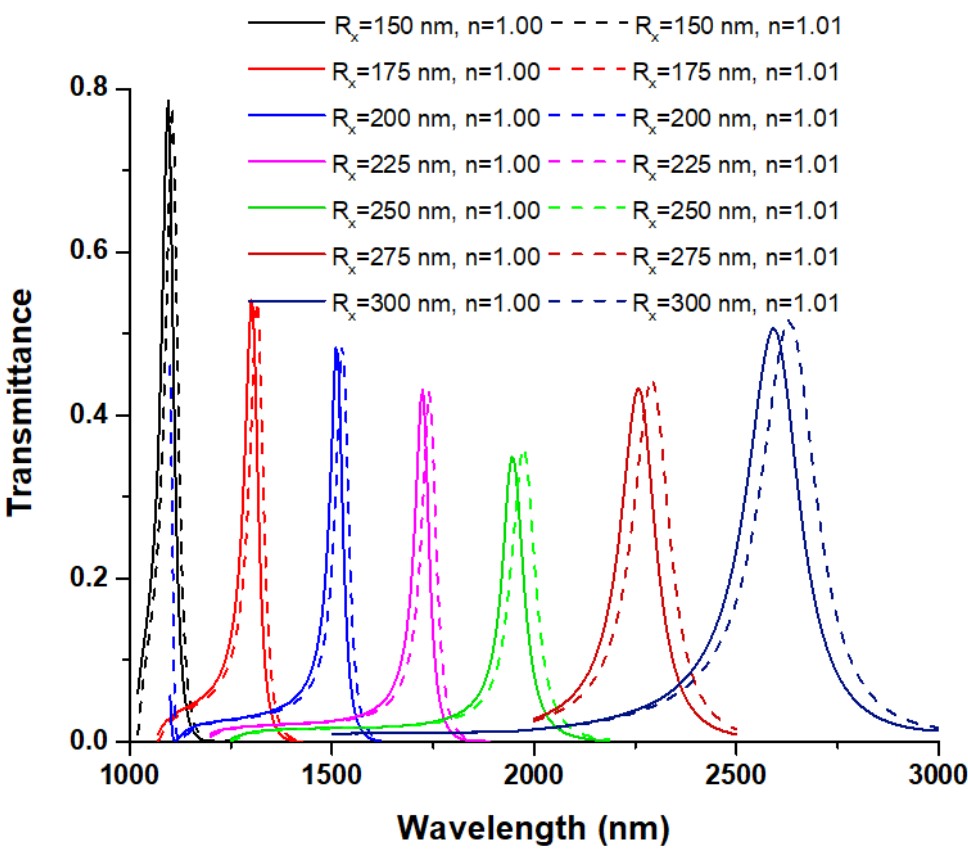

**Figure 6.** Transmittance spectrum of case 4 by changing $R_x$ = (150, 175, 200, 225, 250, 300) nm and the RI (*n*) of surrounding material varying from 1.00 to 1.01.

## 4. Application as a Temperature Sensor

Except for the RI sensing of the above investigation, the proposed case 4 structure can also function as a temperature sensor when the detecting medium is filled in the MIM WG and the elliptical-ring-shaped resonator. We can adopt Equation (3) to inspect the transmittance spectrum versus temperature—the spectrum resonance wavelength changes with the temperature variation in the sensor structure. We employed the structural parameters of Table 2 (i.e., $R_x$ = 200 nm, $R_y$ = 150 nm, w = 50 nm, g = 10 nm, *a* = 40 nm, *b* = 160 nm, *d* = 25 nm, and *r* = 20 nm) because its Fano resonance modes are in the range of visible and infrared regions (see Figure 2a). Figure 7a,b depicts the shift in the transmittance spectrum for temperatures ranging from −100 to 60 °C in the step of 40 °C. Based on Equation (3), the calculated Ethanol RIs for different temperatures are *n* = 1.40776, 1.3920, 1.37642, 1.36048, and 1.34472 for temperature *T* = −100, −60, −20, 20, and 60 °C, correspondingly. The resonance wavelength blueshifts with the increasing temperature because of the decrease of RI with the rising temperature. Figure 8 shows how changing the temperature affects the resonance wavelength. The wavelength shifts are 40 nm, 20 nm, 15 nm, 10 nm, and 10 nm for modes 1–5 when the temperatures vary from −100 °C to 60 °C with the increment of 40 °C, correspondingly. We obtained the temperature sensitivity $S_T$ = 1.0, 0.5, 0.375, 0.25, and 0.25 nm/°C for modes 1–5, respectively. As seen in Figure 8, this proposed case 4 structure also reveals a linear proportion with an increase in the temperature range of −100–60 °C. Note that the higher temperature sensitivity can attain if we use $R_x$ = 300 nm. However, the operating wavelength for mode 1 will shift to mid-infrared, and its FWHM will become more prominent. Besides, if the RI and the temperature are measured at the same time, the temperature may affect the reflectivity of the test. Zhang et al. experimentally inspect RI and the temperature sensor employing the hybrid-sensing probe to measure the temperature and RI simultaneously [64]. Based on

their results, we can measure the RI sensitivity by changing the RI of a solution and keeping the temperature unchanged. Similarly, the temperature sensitivity may be measured in the air environment, avoiding the inherent cross of the RI. The stability and service life of the designed sensor is another important issue and is out of the scope of this work. When the proposed sensor is intended for working in an extreme environment, it requires to consider special technological operations to minimize the deviation of optical properties because of the external factors [65].

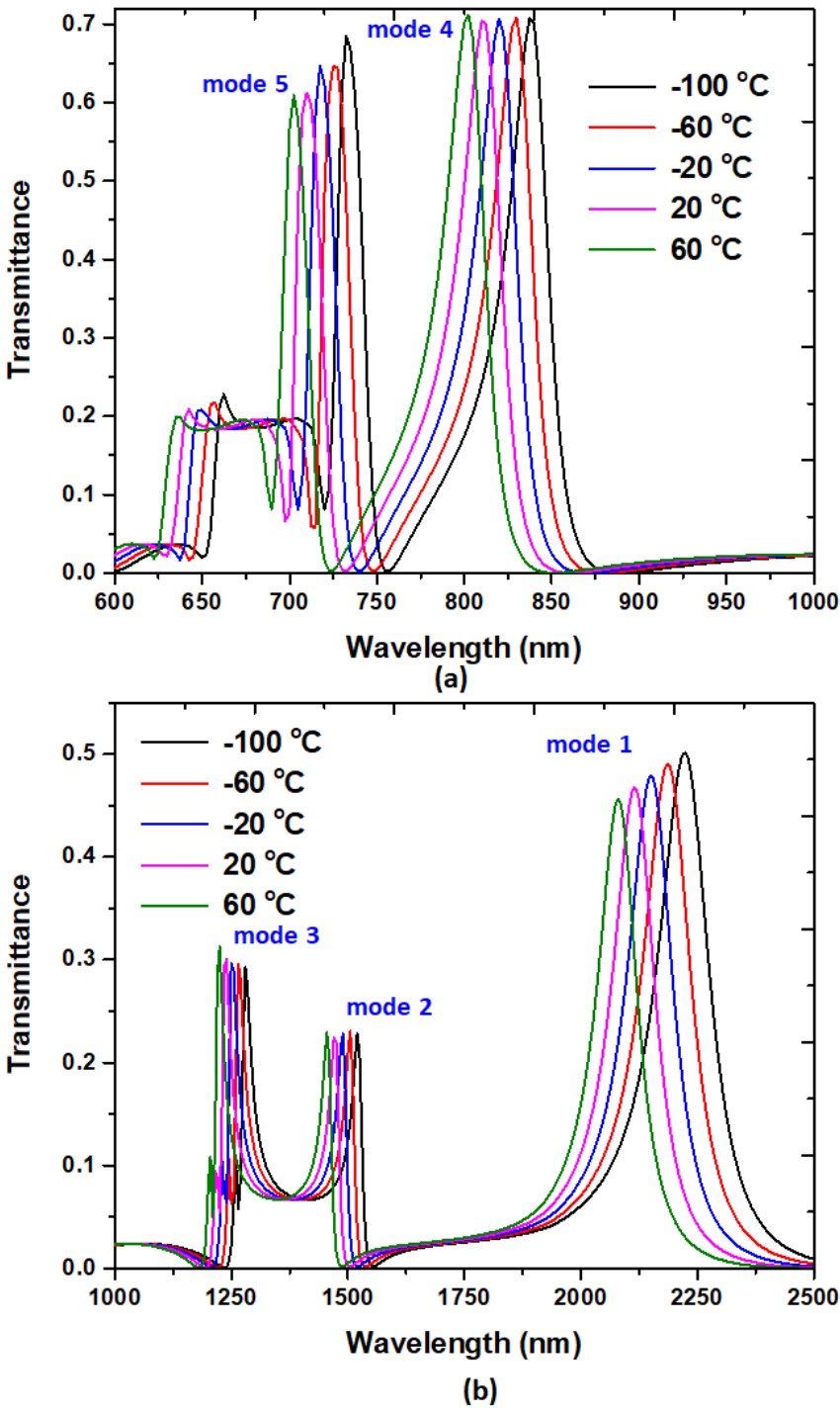

**Figure 7.** Transmittance spectrum of the proposed case 4 structure at various ambient temperatures (*T*) in the wavelength range of (**a**) 600–100 nm and (**b**) 1000–2500 nm. The other parameters remain unchanged from Table 2.

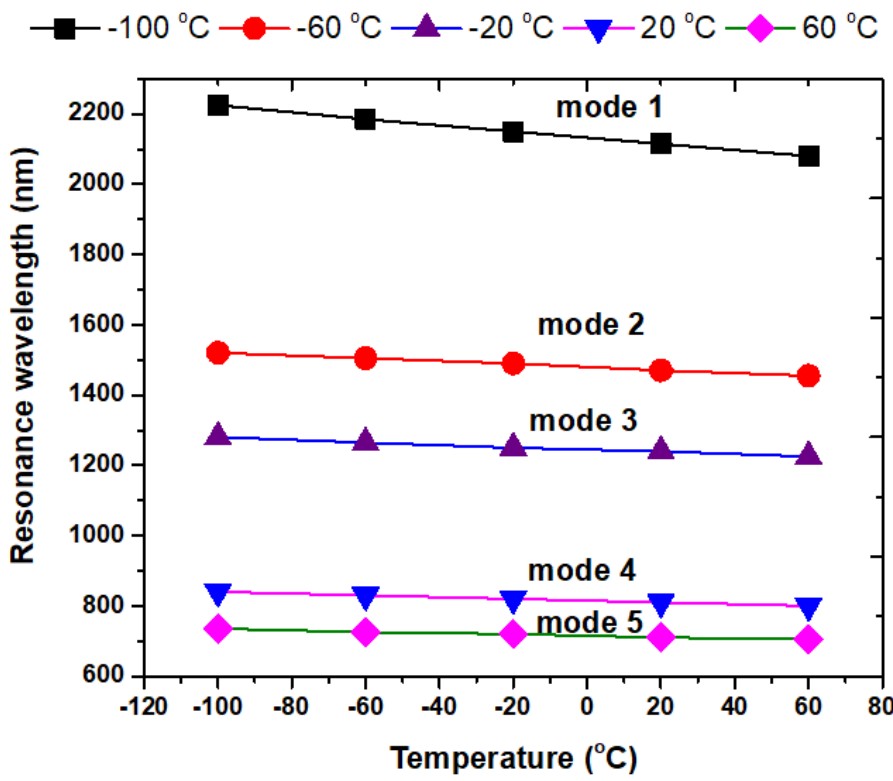

**Figure 8.** The resonance wavelength spectrum of the proposed case 4 structure at various ambient temperatures (*T*), with the other parameters remaining unchanged from Table 2.

The selected electric field (E-field) intensities of case 4 structure at corresponding λresfrom modes 1–5 are shown in Figure 9 at −20 °C, respectively. As seen, the E-field can effectively confine in the resonator because of cavity plasmon resonance. The gap plasmon resonance happened in the gaps between the Ag nanorods and the WG's walls. Besides, some E-fields spread to the stubs, indicating that the stub regions can function as buffer zones to mediate the resonance condition of the elliptical-ring-shaped resonator.

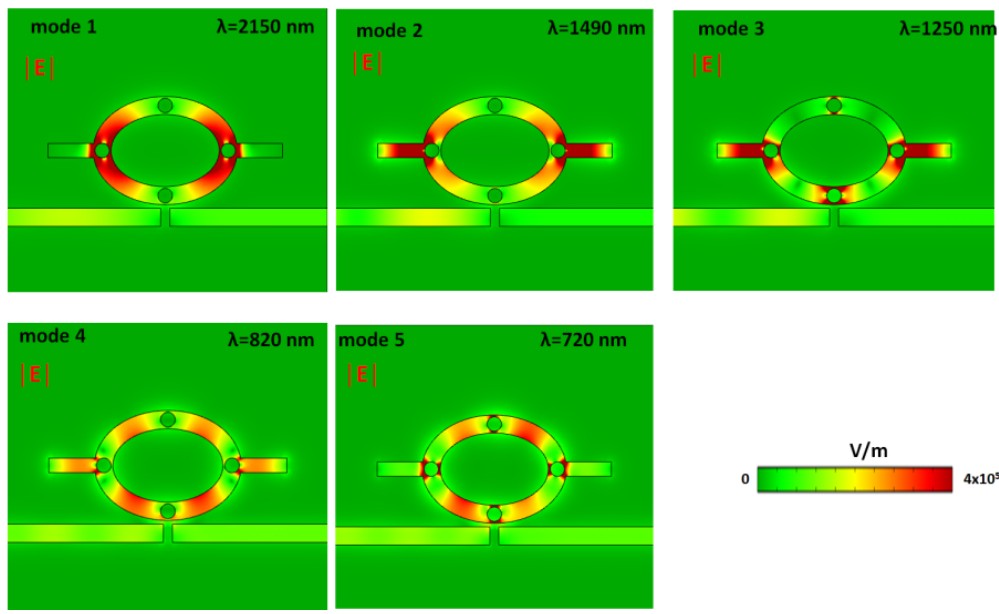

**Figure 9.** Electric field (E-field) intensities of case 4 structure at corresponding $\lambda_{res}$ of modes 1–5.

The proposed case 4 structure can reach high-temperature sensitivity with the same order in the visible and near-infrared wavelength range. To exhibit the excellent performance of the presented case 4 system, Table 4 compares the number of modes, RI sensitivity ($S$), temperature sensitivity ($S_T$), and wavelength range of the designed structure with similar published articles.

**Table 4.** Comparison of the sensitivity ($S$) of the designed system with similar published articles.

| Reference | Mode | Max. RI Sensitivity $S$ (nm/RIU) | Max. Temperature Sensitivity $S_T$ (nm/°C) | Wavelength Range (nm) for $S_T$ | Resonator Size (nm²) |
|---|---|---|---|---|---|
| [66] | 1 | N/A | 0.36 | 1100–1350 | $3.38 \times 10^5$ |
| [67] | 1 | 2320 | 0.84 | 900–3000 | $1.44 \times 10^5$ |
| [68] | 1 | 1326 | 0.53 | 1800–2000 | $1.96 \times 10^5$ |
| [69] | 2 | 2625 | 1.04 | 1000–2200 | $5.76 \times 10^4$ |
| [70] | 4 | N/A | 2.43 | 800–3200 | $5.65 \times 10^5$ |
| This work | 5 | 4500 | 1.00 | 600–2500 | $5.35 \times 10^4$ |

## 5. Conclusions

In summary, we designed a multichannel optical sensor with the function RI and temperature sensing. The proposed case 4 structure consists of two separated MIM WGs side-coupled to an elliptical-ring-shaped resonator. The elliptical-ring-shaped resonator contains two stubs and four Ag nanorods which can mediate the resonance condition and facilitates the light–matter interaction between the MIM WGs and the resonator. We simulated the transmittance spectrum and EM wave distributions using FEM. The results of our proposed case 4 structure indicate that five Fano resonance peaks exist in the transmission spectrum. The maximum sensitivity values can reach 4500 nm/RIU for RI sensing, and the temperature sensitivity can get 1.00 nm/°C. The presented sensor can potentially be useful in nanophotonic sensing applications and is very suitable for the real issues that may need to be considered when the designed sensor is applied in the practical environment, such as food safety inspection and toxic gas concentration.

**Author Contributions:** Conceptualization, investigation, and data curation C.-T.C.C.; methodology, software, writing—review and editing, Y.-F.C.C. All authors have read and agreed to the published version of the manuscript.

**Funding:** This research was funded by the University Research Grant of Universiti Brunei Darussalam, grant number UBD/RSCH/1.9/FICBF(b)/2022/018.

**Institutional Review Board Statement:** Not applicable.

**Informed Consent Statement:** Not applicable.

**Data Availability Statement:** Not applicable.

**Acknowledgments:** The authors thankfully acknowledge the financial support rendered by the Universiti Brunei Darussalam.

**Conflicts of Interest:** The authors declare no conflict of interest.

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
