# Peer review of "Highly Sensitive Multichannel Fano Resonance-Based Plasmonic Sensor for Refractive Index and Temperature Sensing Application"

_photonics, doi:10.3390/photonics10010082_

Round 1

Reviewer 1 Report

The manuscript proposed a susceptible plasmonic sensor for sensing refractive index (RI) and temperature media working in the visible to near-infrared range. Some issues must be clarified before it is accepted for publication in Photonics.

1.     The dimensions of every part of the structure are nanometer order, and the radius of Ag nanorods are 0~22nm, how to manufacture a structure with such small dimensions? And if the structure has been manufactured, how to measure such small dimensions? And how to place and fix such thin Ag nanorods in right positions?

2.     Where is table 2?

3.     In Fig.3,what is the value of d when the value of g increases from 5 to 30nm? And what is the value of g when the value of d increases from 0 to 50nm?

4.     What are the values of b and d when the value of a increases from 10 to 80nm? And what are the value of a and b when the value of d increases from 0 to 40nm?

5.     In Fig.6 and Fig.7, the RI and temperature have been sensed respectively. If the RI and temperature of a media vary at the same time, how to measure RI and temperature simultaneously and independently?

Author Response

Reply to reviewer’s comments

Reviewer 1

Not applicable

The manuscript proposed a susceptible plasmonic sensor for sensing refractive index (RI) and temperature media working in the visible to near-infrared range. Some issues must be clarified before it is accepted for publication in Photonics.

  1. The dimensions of every part of the structure are nanometer order, and the radius of Ag nanorods are 0~22nm, how to manufacture a structure with such small dimensions? And if the structure has been manufactured, how to measure such small dimensions? And how to place and fix such thin Ag nanorods in right positions?

[Reply] Thanks for the comment. This is a very good question. We have explained this question in the revised manuscript. (Line 247-256 on page 7)

  1. 2.Where is table 2?

[Reply] Thanks for indicating the typo. We have corrected it in the revised manuscript. (see line 165 on page 4)

  1. In Fig.3, what is the value of d when the value of g increases from 5 to 30nm? And what is the value of g when the value of d increases from 0 to 50nm?

[Reply] Thanks for the comment. We addressed this question in the revised manuscript (see 195-197 on page 6).

  1. What are the values of b and d when the value of a increases from 10 to 80nm? And what are the value of a and b when the value of d increases from 0 to 40nm?

[Reply] Thanks for the comment. We addressed this question in the revised manuscript (see 218-220 on page 6).

  1. In Fig.6 and Fig.7, the RI and temperature have been sensed respectively. If the RI and temperature of a media vary at the same time, how to measure RI and temperature simultaneously and independently?

[Reply] Thanks for the comment. We addressed this question in the revised manuscript (see 297-302 on page 9).

Reviewer 2 Report

In this paper the authors propose a susceptible multichannel plasmonic sensor for sensing refractive index (RI) and temperature media working in the visible to near-infrared range. The proposed structure's resonator consists of an elliptical-shaped ring with two stubs at two sides and four metal nanorods side-coupled to two separated metal-insulator-metal waveguides. Maximum sensitivity values of the proposed structure can reach 4500 nm/RIU for RI sensing. However, similar work has been reported before in [1]. I personally think that the novelty of this work may be a problem. Moreover, the proposed model is only obtained in the case of simulation, and putting forward such a model without experimental results is a weak contribution for the community. For these reasons, I recommend an objection to the publication of this article in the journal photonics.

 [1] M. R. Rakhshani, “Fano resonances based on plasmonic square resonator with high figure of merits and its application in glucose concentrations sensing,” Opt. Quantum Electron. 51(9), 287 (2019).

Author Response

In this paper the authors propose a susceptible multichannel plasmonic sensor for sensing refractive index (RI) and temperature media working in the visible to near-infrared range. The proposed structure's resonator consists of an e

lliptical-shaped ring with two stubs at two sides and four metal nanorods side-coupled to two separated metal-insulator-metal waveguides. Maximum sensitivity values of the proposed structure can reach 4500 nm/RIU for RI sensing. However, similar work has been reported before in [1]. I personally think that the novelty of this work may be a problem. Moreover, the proposed model is only obtained in the case of simulation, and putting forward such a model without experimental results is a weak contribution for the community. For these reasons, I recommend an objection to the publication of this article in the journal photonics.

 [1] M. R. Rakhshani, “Fano resonances based on plasmonic square resonator with high figure of merits and its application in glucose concentrations sensing,” Opt. Quantum Electron. 51(9), 287 (2019).

[Reply] Thanks for the comment. The key finding of this work is that the existence of four silver nanorod defects in the resonator has a great influence on the sensitivity performance, which provides an additional degree to manipulate the system response in the nanometer scale. The advantage of the proposed structure has a compact size of 5.35×104 nm2, and can offer five channels of Fano resonance modes, which is superior to that of “Opt. Quantum Electron. 51(9), 287 (2019)”, in which can only support one Fano resonance mode and a bigger resonator size of 4.34×105 nm2.

To address the reviewer’s concern, we clarified this question and quoted “Opt. Quantum Electron. 51(9), 287 (2019)” in the revised manuscript (see 76-78 on page 2).

Reviewer 3 Report

Chao and coworkers report on the multichannel Fano resonance-based plasmonic sensor for refractive index and temperature sensing application. the topic is interesting, this paper has solid date, the results presented seems reasonable and convincing. So I recommend that it could be publication after minor consideration.

1. If the refractive index and temperature are measured at the same time, the temperature may affect the reflectivity of the test, so how to avoid signal interference.

2. How is the stability and service life of the design sensor, what is the temperature detection range?

3. It is suggested to discuss some real issues that may need to be considered when the designed sensor is applied in the practical environment. And make a comparsion with other structural sensors.

Author Response

Reviewer 3

Chao and coworkers report on the multichannel Fano resonance-based plasmonic sensor for refractive index and temperature sensing application. the topic is interesting, this paper has solid date, the results presented seems reasonable and convincing. So I recommend that it could be publication after minor consideration.

  1. If the refractive index and temperature are measured at the same time, the temperature may affect the reflectivity of the test, so how to avoid signal interference.

[Reply] Thanks for the comment. This is a very good question. We addressed this question in the revised manuscript (see line 297-302 on page 9).

  1. How is the stability and service life of the design sensor, what is the temperature detection range?

[Reply] Thanks for the comment. We reply “How is the stability and service life of the design sensor” on line 303-307 of page 7, and we addressed “the temperature detection range” on line 127-129 of page 4 in the revised manuscript.

  1. It is suggested to discuss some real issues that may need to be considered when the designed sensor is applied in the practical environment. And make a comparsion with other structural sensors.

[Reply] Thanks for the comment. We addressed this question in the revised manuscript (see 350-352 on page 12). Besides, the comparison with other structural sensors are summarized in Table 4.

Round 2

Reviewer 2 Report

The author has elaborated on its innovation in the paper, and I think this paper is innovative enough to be published after modification.